# Multiplexed Human Gene Expression Analysis Reveals a Central Role of the TLR/mTOR/PPARγ and NFkB Axes in Burn and Inhalation Injury-Induced Changes in Systemic Immunometabolism and Long-Term Patient Outcomes

**DOI:** 10.3390/ijms23169418

**Published:** 2022-08-20

**Authors:** Cressida Mahung, Shannon M. Wallet, Jordan E. Jacobs, Laura Y. Zhou, Haibo Zhou, Bruce A. Cairns, Robert Maile

**Affiliations:** 1North Carolina Jaycee Burn Center, Department of Surgery, Chapel Hill, NC 27514, USA; 2Division of Oral and Craniofacial Health Sciences Adams School of Dentistry, University of North Carolina School of Medicine, Chapel Hill, NC 27599, USA; 3Department of Microbiology and Immunology, University of North Carolina School of Medicine, Chapel Hill, NC 27599, USA; 4Department of Biostatistics, University of North Carolina School of Medicine, Chapel Hill, NC 27599, USA; 5Curriculum in Toxicology and Environmental Medicine, University of North Carolina School of Medicine, 8031 Burnett Womack, Chapel Hill, NC 27599, USA

**Keywords:** burn injury, inhalation injury, surgical outcomes, trauma

## Abstract

Burn patients are subject to significant acute immune and metabolic dysfunction. Concomitant inhalation injury increases mortality by 20%. In order to identify specific immune and metabolic signaling pathways in burn (B), inhalation (I), and combined burn-inhalation (BI) injury, unbiased nanoString multiplex technology was used to investigate gene expression within peripheral blood mononuclear cells (PBMCs) from burn patients, with and without inhalation injury. PBMCs were collected from 36 injured patients and 12 healthy, non-burned controls within 72 h of injury. mRNA was isolated and hybridized with probes for 1342 genes related to general immunology and cellular metabolism. From these specific gene patterns, specific cellular perturbations and signaling pathways were inferred using robust bioinformatic tools. In both B and BI injuries, elements of mTOR, PPARγ, TLR, and NF-kB signaling pathways were significantly altered within PBMC after injury compared to PBMC from the healthy control group. Using linear regression modeling, (1) DEPTOR, LAMTOR5, PPARγ, and RPTOR significantly correlated with patient BMI; (2) RPTOR significantly correlated with patient length of stay, and (3) MRC1 significantly correlated with the eventual risk of patient mortality. Identification of mediators of this immunometabolic response that can act as biomarkers and/or therapeutic targets could ultimately aid the management of burn patients.

## 1. Introduction

Severe burn injury is one of the most devastating forms of trauma, with mortality rates reaching up to 12% even in specialized burn centers. In the United States, approximately 450,000 burn injuries occur each year that require medical treatment, with nearly 3500 deaths [1,2]. Mortality due to a severe burn injury is primarily due to complications, including organ failure, pneumonia, and other infections (reviewed in [3]). This is in part because burn patients are subject to significant immune and metabolic derangements, beginning with an initial genomic storm [3,4] and concurrent systemic inflammatory response, followed by a compensatory anti-inflammatory response [5] and hypermetabolic response, which can last for years (reviewed in [3]). Identification of mediators of this dysregulated immunometabolic response is crucial for the management of the compromised immune system and catabolic alterations faced by burn patients. Of the multiple influences on morbidity and mortality in burn patients, inhalation injury is among the most significant [6]. Combined burn and inhalation injury (BI) occurs in 5–30% of all burn patients and is characterized by epithelial denudation, elevated leukocyte (neutrophil and macrophage) activity in the lung, with enhanced local and systemic inflammation. Together, these lead to an increased morbidity and mortality, including increased lung damage and bacterial infections.

It is generally accepted that burn injury results in a biphasic systemic immune dysfunction. The acute phase (0–72 h post-injury) is referred to as burn shock or systemic inflammatory response syndrome (SIRS) and can lead to epithelial and endothelial barrier dysfunction and multiple organ failure in the early time points after injury [7]. For most patients that survive the acute phase, it becomes clinically obvious that they have entered a late/chronic phase of immune dysfunction, which is often referred to as the compensatory anti-inflammatory response syndrome (CARS). This phase is clinically associated with an increased susceptibility to infection [8,9]. These phases have great heterogeneity, and thus are more clearly defined as a mixed agonist response syndrome (MARS) [10].

It has been shown that burn injury generates numerous inflammatory stimuli, including damage-associated molecular patterns (DAMPs), such as hyaluronic acid (HA) and double-stranded DNA (dsDNA), to which resident innate immune cells (neutrophils and macrophages) and non-canonical immune responder cells (such as pulmonary epithelial cells) bind through pattern recognition receptors (PRRs), including toll-like receptors (TLRs; reviewed in [11]). This results in cellular activation and induction of the mammalian/mechanistic target of rapamycin (mTOR)/peroxisome proliferator-activated receptor γ (PPARγ) axis. mTOR activation drives the execution of metabolic cellular programming and inflammatory functions, both hallmarks of MARS [12]. Specifically, mTOR activation leads to upregulation of the negative regulator of inflammation, PPARγ, which in turn represses immune responses [13]. To this end, in addition to the immunological complications observed, patients experience severe metabolic dysregulation, characterized by elevated basal metabolic rates and a paradoxical commingling of hyperglycemia and hyperinsulinemia [14]. A key result of metabolic and immune re-programming by PPARγ, is the activation of anti-inflammatory signals, such as the cytokine interleukin-10 (IL-10), promotion of restorative processes, such as the polarization of macrophages and neutrophils into alternatively activated type-2 anti-inflammatory macrophage (M2) and neutrophil (N2) populations, and subsequent susceptibility to infection. Indeed, early pulmonary and systemic expression of the mTOR/PPARγ regulates both the anti-inflammatory cytokine IL-10 and the metabolic enzyme arginase 1 (ARG1), both of which are important predictors of immune suppression in burn patients [15,16]. Thus, it is plausible that the immune and metabolic system dysfunction resulting from DAMP-induced activation of the TLR/mTOR/PPARγ pathway is responsible for poor patient outcomes. Although previous studies have explored immunological dysfunction during burn injury, no study has identified biomarkers, which can inform the clinical decision-making to assess patients’ immune status, contributing to poor patient outcomes [17]. Here, an unbiased survey of immune and metabolic gene expression was performed within peripheral blood monocytic cells (PBMCs) obtained early after injury (<48 h) from burn patients, with and without inhalation injury. These data demonstrate that there is a significant alteration in immune and metabolic genes, and thus signaling pathways following injury, which also have the potential to be used as biomarkers and/or predictors to determine patients’ risk for poor outcomes following burn injury, inhalation injury, and/or combined burn/inhalation injury.

## 2. Results

### 2.1. Burn Injury Patient Characteristics and Demographics

The overall study design is summarized in Figure 1. In brief, semi-untargeted gene expression analysis was performed on PBMC collected within 48 h of injury from 36 burn patients and 12 healthy patients for the control group (Table 1). Burn injury patients were stratified into one of three groups: those that had burn injury alone (B; n = 19), inhalation injury alone (I; *n* = 3), and combined burn and inhalation injury (BI; n = 14).

### 2.2. Burn Injury, Inhalation, and BI Injury Induces Significant Alterations in Peripheral Immunometabolic Gene Expression Changes Compared to Healthy

It was hypothesized that systemic immunologic and metabolic dysfunction after burn (B) injury is driven at least in part due to the mTOR/PPARγ signaling axis leading to acute re-programming of systemic immune cell metabolism. As highlighted in Introduction, the presence of inhalation (I) injury is known to significantly reduce patient survival by up to 20% after burn injury. Therefore, mRNA was isolated from PBMC samples collected within 48 h of B, I, or BI injury. mRNA was interrogated for the expression of 1342 genes involved in the execution and regulation of pathways associated with immunology and metabolism using nanoString technology.

Analysis of the PBMC mRNA expression (Figure 2 and Figure 3, and Appendix A), revealed that B, I, and BI injuries promote significant changes in immune gene mRNA expression. Figure 2 demonstrates a (1) relative expression as depicted by heat maps (Figure 2A) and a significant fold-change as depicted by volcano plots (Figure 2B). Compared to healthy controls, 582 genes (347 increased, 235 decreased; Log2 fold-change −2.90–4.63) were significantly affected by B injuries (*p* < 0.05), 259 genes (164 increased, 95 decreased; Log2 fold-change −1.92–4.77) were significantly affected by I injuries, and 525 genes (317 increased, 208 decreased; Log2 fold-change −2.41–4.97) were significantly affected by BI injuries. As summarized in Figure 3, there were 263 genes, which are commonly affected by B and I injuries (146 increased, 90 decreased), 463, which are commonly affected by B and BI injuries (282 increased, 181 decreased), 214, which are commonly affected by I and BI injuries (129 increased, 85 decreased), and 210 genes, which are commonly affected by B, I, and BI injuries (125 increased, 85 decreased). Gene lists for each comparison of each injury modality *versus* the corresponding healthy component are provided in Appendix A. In conclusion, transcriptome analysis of the systemic PBMC response did demonstrate significant injury modality-specific immune and metabolic gene expression changes compared to healthy individuals.

### 2.3. Burn Injury Alone, Inhalation Injury Alone, and Combined BI Injury Induce Significant Alterations in Peripheral Immune Cell Composition Compared to Healthy Controls

NanoString analysis was also used to profile the composition of immune cell types, utilizing pre-defined sets of genes to de-convolute relative frequencies of immune cell types. This method has been verified to correlate well with other methods using clinical samples, such as flow cytometry [18]. Based on PBMC gene expression, the presence of cytotoxic cells, CD45+ cells (pan-leukocyte marker), macrophages, T cells, natural-killer (NK), T helper 1 (Th1) T cells, and CD8 T cells were calculated (Figure 4). Regardless of burn injury or health, there were equivalent numbers of CD45+ cells, demonstrating that there are no gross changes in immune cell numbers after injury. PBMC samples from both B and BI injuries, compared to PBMC from healthy controls, presented a significant decrease in the abundance of cytotoxic cells, T cells, NK, Th1, and CD8 cells (*p* < 0.05), and a significant increase in the abundance of macrophages (*p* < 0.05). PBMC from inhalation injury alone did not present with any significant changes in abundance of the immune cell types quantified when compared to healthy controls. In conclusion, these data suggest that burn injury, either in isolation or combined with inhalation injury, is the main driver of changes in systemic immune cell populations.

### 2.4. Burn Injury Alone, Inhalation Injury Alone and Combined BI Injury Promote Significantly Different Pathway Scores within Immunometabolic Pathways

In order to identify the specific immune and metabolic pathways significantly affected by each injury modality, principal component analysis (PCA) was used to generate pathway scores (PS). PS are quantitative, and based on the individual gene expression levels, all the measured genes within a specific pathway were derived [19]. Figure 5 shows the PS for gene expression profiles from PBMC isolated from healthy individuals and all burn injury categories. Burn injury alone, I and BI injuries significantly invoked pathways involved in adhesion, chemokine, cytokines, and transporter functions, and significantly downregulated antigen processing, cell function, cytotoxicity, NK cell functions, and a regulatory pathway. Burn injury alone and combined BI, but not I injury alone, significantly downregulate pathways associated with cell-cycle, pathogen defense, and TLR signaling. In conclusion, these findings suggest that all three injury modalities studied independently induce acute systemic changes in a wide range of immune and metabolic pathways.

### 2.5. Burn, Inhalation, and BI Injuries Induce Significant Alterations in Peripheral Immunometabolic Canonical Immune and Metabolic Signaling Pathways Compared to Healthy Controls

While PS can indicate the high-level directionality of generalized pathways, ingenuity pathway analysis (IPA) was utilized to identify causal networks. IPA uses transcriptome data to identify cause–effect relationships, and reports Z-scores for known canonical pathways. Pathways are ranked based on their Z-Score, with a high Z-Score indicating that the pathway is more affected than expected based on the overall dataset [20]. Figure 6 shows immune and metabolic pathways that were significantly affected in burn patients compared to healthy controls (*p* < 0.05). The mTOR, PPARγ, TLR, and nuclear factor kB (NF-kB) signaling pathways were identified as key immunometabolic pathways altered in PBMCs after injury (Z-score > 2, *p* < 0.05). PPARγ signaling presented decreased activation after injury, while TLR and NF-kB signaling demonstrated increased activation.

Causal molecular pathways were constructed using differential gene expression within each injury type (B, I, or BI) versus healthy. This analysis focused on the mTOR (Appendix A), TLR (Appendix A), and NFkb (Appendix A) pathways. Figure 7 shows the key elements within each of these pathways that burn injury alone affect as it relates to healthy controls. Each figure visualizes the positive (red) or negative (green) fold-changes. Specifically, Figure 7 and Appendix A demonstrate, following B and BI injuries, that both mTOR/PPARγ and TLR/NF-kB signaling pathways were significantly altered, where the following components of each of these signaling pathways were consistently and significantly altered; (1) an increase in TLR1, 2, 4, 5, IL-1R, and tumor necrosis factor receptor (TNFR) expression, (2) reduction in specific mTOR components of regulatory associated protein of MTOR (RPTOR) and protein observed with Rictor-1 (PROTOR), (3) an increase in the mTOR components late endosomal/lysosomal adaptor, MAPK and MTOR Activator 2 (LAMTOR2) and late endosomal/lysosomal adaptor, MAPK and MTOR Activator 5 (LAMTOR5), (4) an increase in PPARγ expression (which acts to downregulate mTOR activation), (5) an increase in the expression of IkB (which inactivates NFκB transcription factor), (6) an increase in anti-inflammatory IL-10 and ARG1, and (7) an increase in mannose receptor C-type 1 (MRC1) expression (a key macrophage lectin sensor known to be regulated by the mTOR pathway and expressed by polarized anti-inflammatory M2 macrophages).

Considering each pathway in turn (Appendix A), at a greater resolution, it is clear that there are significant alterations in key signaling pathways between injury and health, as well as between injury modalities. Within the mTOR signaling pathway, burn injury drives upregulation of the Proline-Rich Akt Substrate of 40kDa (PRAS40) and RPTOR. The impact of BI is very similar to that of burn alone, although a significantly less impact on Hypoxia-inducible factor (HIF1) was observed. While the impact of inhalation injury alone on immunometabolic gene expression was similarly significant, there was less systemic mTOR involvement versus B or BI. Within the TLR signaling pathway, in burn injury versus healthy controls, there were significant alterations in the expression of innate sensing molecules; TLRs 1, 2, 4 (including CD14), 5, 8, Myeloid Differentiation factor 2 (MD-2), Myeloid differentiation primary response 88 (MyD88), and Lymphotoxin Beta Receptor (LTBR) were all significantly elevated. TLRs 3, 7, 9 and 10 were not significantly impacted. Single Ig IL-1-Related Receptor (SIGIRR) was significantly down regulated. PPARγ was significantly increased in burn patients *versus* healthy controls. When evaluating the impact of inhalation injury, there was a similar pattern of expression although CD14 and SIGRR were less impacted, and PPARγ was not altered. When evaluating the impact of combined BI injury, a clear synergistic effect these injuries was observed, with PBMCs from BI patients, *versus* healthy controls, having enhanced expression of the same innate receptors compared to PBMCs from burn injured patients alone. In addition, TLR10 was significantly downregulated in PBMCs from these patients when compared to healthy PBMCs, unlike what was observed in PBMC from burn or inhalation patients. These patterns are also borne out in the NFκB pathways, with PBMC-expression showing B, I, and BI impacting many key elements, including the alternate pathway of NFκB activation. Of note, B and BI injuries increase PBMC expression of RELB Proto-Oncogene, NF-KB Subunit (RelB) in contrast to inhalation injury itself. B and I injuries also induce key inhibitory alpha and gamma molecules of the IκB family (NEMO). For combined BI injuries, induction of Inhibitor of NFκB Kinase subunit β (IKKβ) rather than IKKα was observed. In conclusion, this causal pathway analysis defined unique and overlapping impacts of injury on intrinsic mTOR and NFkB-based regulatory feedback mechanisms that occur acutely after injury.

### 2.6. Linear Regression Modeling Revealed Significant Association between PBMC DEPTOR, LAMTOR5, MRC1, PPARγ, and RPTOR Gene Levels and Patient Outcomes 

In order to guide patient care and develop a predictive model, allowing physicians to provide anticipatory care improving patient outcomes, multivariate regression analyses were used to test for associations between key genes of these pathways expressed early after injury and clinical patient outcomes as previously described [15,16,21]. Due to their significant crosstalk between immune and metabolic functions and altered expression (Figure 7 and Appendix A), it was hypothesized that key mTOR and PPARγ molecules would associate significantly with patient characteristics and clinical outcomes. Linear regression analysis focused on the following PBMC-expressed genes: DEP-Domain containing MTOR interacting protein (DEPTOR), LAMTOR5, MRC1, PPARγ, RPTOR, macrophage receptor with collagenous structure (MARCO), mTOR, and MyD88. In addition, the following confounders were included into the model: burn severity (percent total body surface area; TBSA), age, body mass index (BMI), race, sex, length of stay (LOS), inhalation injury, expiration, development of acute lung injury (ALI), infection of blood or bronchoalveolar lavage (BAL), and required a ventilator. Using these parameters, a following regression model was utilized (Equation (1)), where *Y* = outcome variable of interest.
(1)$$\begin{array}{ll} \log_{2}\left(Y+1\right)= & \beta_{0}+\beta_{1}Age+\beta_{2}+\beta_{3}+\beta_{4}BMI+\beta_{5}pct_{TBSA}+\beta_{6}+\beta_{7}LOS+\beta_{8}Expired\\ & \;+\beta_{9}Develop\_ALI+\beta_{10}Infection\_Blood\_BAL+\beta_{11}Required\_Ventilator \end{array}$$

The covariates were defined as race = 1 if race is “Non-white”, else 0; sex = 1 if male, else 0; inhalation injury = 1 if yes, else 0; expired = 1 if yes, else 0; develop ALI = 1 if yes, else 0; and infection BB = 1 if “Infection of Blood or BAL”, else 0. Each of the targeted genes and/or their ratios were tested for association with each of the clinical outcomes taking the confounders into consideration. To test whether there was a significant change in the outcome for one unit change in the i−th continuous covariate or between covariate levels for a categorical covariate, holding all other covariates constant, it was tested whether regression coefficient $\hat{\beta_{i}}=0$. Appendix A shows the quantile-quantile (Q-Q) plots with summative data from all injury modalities which confirm theoretical and sample quantiles have the same distribution and “goodness of fit” of the developed model for individual DEPTOR, LAMTOR5, MRC1, PPARγ, RPTOR, MARCO, mTOR, and MyD88 values. Table 2 lists the parameter estimates, standard error, and *p*-values for the covariates in the model, based on the Wald test. Taking each gene in turn, DEPTOR gene levels are significantly associated with BMI ($\hat{\beta }$, the estimated β coefficient, =0.013, *p* = 0.049) and the presence of inhalation injury ($\hat{\beta }$ = 0.343, *p* = 0.099). LAMTOR5 ($\hat{\beta }$ = 0.026, *p* = 0.012), RPTOR ($\hat{\beta }$ = −0.23, *p* = 0.019), and PPARγ ($\hat{\beta }$ = 0.050, *p* = 0.024) were also significantly associated with BMI. RPTOR was also associated with patient death ($\hat{\beta }$ = −0.652, *p* = 0.075), MRC1 was significantly associated with patient LOS ($\hat{\beta }$ = 0.006, *p* = 0.036) and risk of death ($\hat{\beta }$ = 1.803, *p* = 0.048). However, MARCO, mTOR, and MyD88 were not significantly associated with any clinical covariates. In conclusion, certain genes with acutely altered expression compared to healthy patients in PBMC after injury may indeed act as biomarkers indicating poor patient outcomes.

## 3. Discussion

Major traumatic injuries, such as burn injuries, result in a systemic inflammatory response syndrome characterized by dramatic alterations within the immune system [3]. Burn injuries are one of the most serious forms of trauma and are associated with high morbidity and mortality, particularly when associated with an inhalation injury. Clinical outcomes from burns are inextricably linked with immune function, with patients defined as having a tissue-damaging pro-hyperinflammatory response early after injury. Patients surviving the initial clinical phase often succumb to systemic and pulmonary hospital-acquired infections (HAI), such as *Pseudomonas aeruginosa* [22], with a more chronic immunosuppressed syndrome. As discussed in the introduction, we, and others, have shown that burn injury generates numerous systemic and local inflammatory stimuli, including DAMPs released from dying tissue and cells, and engages pattern recognition receptors, such as TLR. This results in the activation of innate immune cells (e.g., macrophages and neutrophils) and stromal cells (e.g., pulmonary epithelium), and the activation of several signaling pathways, including those of mTOR and NFκB. Together, these axes drive the execution of metabolic cellular programming and inflammatory functions, causing further local and systemic damage and cytokine-storm (“burn shock”) [12]. A molecular model of the acute burn injury-induced imbalance within the mTOR/PPARγ and NFκB/IκB immunoregulatory axes early after an injury that sets up the chronic clinical susceptibility to infection is presented in Figure 8.

Our immune system has evolved to self-regulate and protect the host from tissue damage through the induction of anti-inflammatory, wound healing, and re-modeling pathways. For instance, mTOR activation leads to the rapid upregulation of the negative regulator of inflammation, PPARγ, which in turn represses mTOR-activation [13], while also promoting anti-inflammatory pathways [23]. Similarly, NFκB-mediated transcription induces the upregulation of its negative regulator, IκB. While PPARγ also regulates the NFκB pathway through both physical interactions with NFκB, preventing its nuclear translocation, and active transcription of IκB [24] (Figure 8).

While modern critical care can resuscitate severely burned patients, these intrinsic regulatory feedback mechanisms cannot sufficiently counter and recover from the high degree of tissue damage and “burn shock” that occurs in this population. In support of this, it has been demonstrated that polarized anti-inflammatory neutrophils and macrophages arise rapidly after burn and BI injuries, and they are ineffective at dealing with repeated inflammatory insults, including bacterial infections [25]. We have previously described a set of systemic (PMBC) immune polarization genes that correlate strongly with patient outcomes [16]. Here we present new data that delineates a group of immunoregulatory genes with systemic expression levels that are significantly altered early (<48 h) after burn injury and complement our previously described immune polarization genes.

This work has identified PPARγ and IκB as two of the most highly induced members after injury. We have observed (1) an increase in TLR1, 2, 4, 5, IL-1R, and TNFR expression (as predicted by us and other studies, representing engagement of innate sensing molecules and cytokine receptors), (2) reduction in specific mTOR components RPTOR and PROTOR, (3) an increase in mTOR components LAMTOR2 and LAMTOR5, (4) an increase in PPARγ expression (which acts to downregulate mTOR activation), (5) an increase in expression of IkB (which inactivates NF-κB transcription factor), (6) an increase in anti-inflammatory IL-10 and ARG1 (also predicted in our earlier study) [16], and (7) increase in MRC1 expression (a key macrophage lectin sensor known to be regulated by the mTOR pathway and expressed by polarized anti-inflammatory “M2” macrophages). In addition, with regard to the mTOR axis, burn injury drives upregulation of PRAS40 and RPTOR. PRAS40 acts as an intersection of the Akt- and mTOR-mediated signaling pathways.

When the association of these genes with patient characteristics and outcomes was investigated, we found that PBMC DEPTOR, LAMTOR5, PPARγ, and RPTOR fold-changes significantly correlated with patient BMI. It has been previously shown that normal-weight and mildly obese patients respond better to critical illness when compared with morbidly obese patients [26,27,28], and these data suggest a mechanism to more fully investigate this association. RPTOR was also associated with patient death. MRC1, a key macrophage pathogen sensor regulated by the mTOR pathway, was significantly correlated with the eventual risk of patient mortality and length of stay.

The novelty of this study lies in multiple areas. Firstly, the identification of signaling pathways invoked after an injury is vital. No therapeutic agent exists to return the burn patient’s immune system to homeostasis. It is well accepted that immunometabolic dysfunction is initiated rapidly following burn injury, with the concept that burn care provided as early as possible is important for survival, and resolution of dysfunction during this phase of injury has the potential to improve long-term clinical outcomes. Dogma in the field suggests that inhibition of pro-inflammatory responses and/or induction of anti-inflammatory responses during this time would reverse the immune dysfunction following burn injury. Unfortunately, attempts to modulate these phases of immune dysfunction with cytokine/anti-cytokine therapy have largely been unsuccessful [29], suggesting the additional multi-modal immunological interactions also play a role in driving clinical disease. Indeed, these data indicate that interruptions in the mTOR and NFkB pathways prevent the inflammatory sequelae associated with burn injury-induced immune dysfunction and partially restore bacterial resistance. With that said, our data also suggest that regulatory feedback loops are also induced but are either dysfunctional or ineffective, such that acute and long-term pathology still ensues. Thus, it is plausible that enhancing these regulatory feedback loops would promote immune homeostasis. On the other hand, there are data to suggest that early immune regulation leads to ineffective immune cell function in the long run, such that early inhibition of inappropriately induced regulatory feedback loops would be beneficial in restoring immune homeostasis. We are currently investigating the potential benefit of enhancing (agonizing) or correcting (antagonizing) the early invocation of these regulatory gene products in mouse models.

Secondly, no patient clinical outcome prediction exists. Although it is evident that the immune dysfunction induced by burn injury correlates with short- and long-term patient outcomes, there is no practical model with a predictive value that accurately reflects the underlying degree of immune insult and significantly correlates with the degree of compromise [17]. In addition, although multi-center bio-informatic studies have described the genomic storm after injury and initial attempts have been made [4,30,31], a profile of immune gene expression from burn patients that can be transformed into an “immune suppression index” has not been translated to the clinic, nor have these genomic markers been used to inform clinical care. Using samples from burn injury and BI-injury patient cohorts in our single high-volume burn center, we have already described a minimal set of genes that are associated strongly with patient outcomes. Specifically, using PBMCs collected <48 h from admission, we previously defined four genes (ARG1, IL10, NOS2, and IL12) that when aggregated into a ratio of (ARG1 + IL-10)/(NOS2 + IL-12), their correlation with patient outcomes was most significant whereby outcomes include lung injury, infection, autograft failure, and death [16]. The data provided here provide novel foundational data towards a more inclusive model of immune and metabolic dysfunction, and provides further targets that can be utilized to define, refine, and validate rapid blood-based “immune suppression index” point-of-care technologies. It is also clear that inhalation injury has a very significant systemic immunometabolic effect, which may have been underestimated, but less systemic mTOR involvement versus burn alone or BI injury.

This study was limited by the nature of a prospective case-control study, with a modest sample size of 36 burn patients and 12 control patients. This was further hindered by the large number of covariates inherent to a trauma population, and while statistical analysis deemed this sufficient for the identification of genes and pathways for follow-up analysis, a more robust and multi-center sample size is needed to validate the greater number of genes identified and a deeper probe into the effect of covariates. Regardless, our gene expression data and subsequent pathway analysis were high yields and the bio-markers described can lend themselves to a simple, rapid diagnostic assay that can be quickly validated and implemented into clinical practice.

## 4. Materials and Methods

### 4.1. Participants

Burn patients admitted to the North Carolina Jaycee Burn Center were recruited into an IRB-approved study. Patients received standard care, and care was not affected by study participation. Patients were not excluded from the study based on burn size, inhalation injury, or other factors, including age, race, or substance use prior to injury. Patients were followed until discharge or expiration (Figure 1). A total of 36 burn patients were included in our case cohort, and 12 never-burned healthy volunteers, nor currently on immunosuppressive therapy, were enrolled as a control cohort (age, 24.8 ± 3.7 y, 17% female; n = 19, age, 26.1 ± 4 y, 53% female). “Strengthening the Reporting of Observational Studies in Epidemiology (STROBE)” guidelines for reporting observational studies were followed [32].

### 4.2. Isolation of Peripheral Blood Mononuclear Cells

Two 8 mL whole blood samples from patients with <48 h post-burn injury were collected by peripheral venipuncture into acid-citrate-dextrose (ACD) solution A. Samples were transferred to mononuclear cell preparation tubes (CPT) and spun at 1750× *g* for 30 min. The cellular fraction was suspended in phosphate-buffered saline (PBS) without calcium or magnesium and centrifuged for 10 min at 300× *g.* The supernatant was decanted, and the cell pellet was re-suspended in 5 mL of ammonium-chloride-potassium (ACK) lysis buffer and incubated at room temperature for 2 min to remove erythrocyte contamination. Cells were washed three times in PBS, counted, and 1 × 10^6^ of peripheral blood mononuclear cells (PBMC) were re-suspended in RLT lysis buffer for total RNA isolation.

### 4.3. RNA Extraction and nanoString Analysis

Total RNA was isolated using the RNAeasy Mini-RNA extraction kit with on-column DNAase digestion according to the manufacturer’s instructions. NanoString hybridization was performed on the nCounter system using the nanoString human immunology and metabolic gene panels (Human Immune Profiling PanCancer CodeSet; 770 genes, and the Human Metabolic Gene CodeSet; 768 genes). This “next generation” multiplex technique is an amplification-free technology that measures gene expression by counting mRNA molecules directly [33] with barcoding to allow multiplex analysis of multiple genes in the same sample.

Briefly, a total of 20 ng mRNA from each sample was hybridized to report captured probe pairs (CodeSets) at 65 °C for 18 hours. After this solution-phase hybridization, the nCounter prep station was used to remove excess probes, align the probe/target complexes, and immobilize these complexes in the nCounter cartridge. The nCounter cartridge was then placed in a digital analyzer for image acquisition and data processing. Fluorescent color codes designating mRNA targets of interest were directly imaged on the surface of the cartridge. The expression level of each gene was measured by counting the number of times the color-coded barcode for that gene was detected, and the barcode counts were tabulated. Background thresholding and normalization were performed using 6 positive control probes and 13 housekeeping genes (POLR2A, SDHA, ABCF1, TBP, EEF1G, RPL19, PPIA, TUBB, HPRT1, OAZ1, GUSB, POLR1B, and GAPDH) through the nSolver v.4 analysis program. Differential-expression volcano plots and heat maps were generated using logarithmically transformed fold changes of averaged-normalized counts for each cell population using healthy control samples as reference. Differential-pathway scores were quantified using modules. Differential-expression was then analyzed using ingenuity pathway analysis (IPA) to identify pathways which were significantly represented by the differential-gene expression.

### 4.4. Immune Cell Deconvolution Analysis

Within the human immunology codeset, there are gene annotation data which delineates 24 immune cell types (aDC, B-cell, CD8 T-cell, Cytotoxic cell, DC, Eosinophils, iDC, macrophages, mast cell, neutrophils, NK CD56^bright^ cell, NK CD56^dim^ cell, NK cell, pDC, T helper cell, T cell, T_cm_, T_em_, TFH, Tγδ, Th, Th17, Th2 cell, and T_reg_ cells). NanoString expression values were log2 transformed and followed by quantile normalization. The immune cell type scores were then calculated by taking the mean of the normalized/transformed expression values of genes defined in the corresponding nanoString gene signature (log2 mean). This was performed within nSolver v.4 Advanced Analysis.

### 4.5. Statistical Analysis

Analysis was conducted after data normality was met using D’Agostino–Pearson. Continuous variables were compared using Mann–Whitney or Kruskal–Wallis with Dunn’s multiple comparisons. Correlation was performed with Spearman rank. Analysis was performed using GraphPad Prism v9.0. For nanoString, a negative binomial mixture model, a simplified negative binomial model, or a log-linear model was used depending on each gene’s mean expression compared to the background threshold. Multiple testing corrections were performed using the method of Benjamini–Yekutieli. Causal network analysis was performed using IPA. Linear regression was used to evaluate the associations between gene transcripts and burn injury clinical outcomes, after adjusting for important confounding variables as we have described previously [15,21]. Log transformation of the clinical variables was used to ensure the model assumptions were met through quantile-quantile (QQ)-plots and other criteria. The linear models were fit in R v3.6.1.

With regards to the sensitivity, error, and robustness of the transcriptome analysis method, nanoString has been well characterized by other studies (for example, [34]) as having extremely high reproducibility (R^2^ > 0.99 for 100 ng input), sensitivity (probe counts between 100 and 500 detected and quantified), and robustness with clinical samples. Variability has also been studied extensively with nanoString, and is highest for the lowest expressed genes (<1 SD), but reduced with an increasing number of field-of-views captured. Cartridge lane-to-lane variability and slot-to-slot variability do not introduce significant bias to the data (for example, [35]).

## 5. Conclusions

This study, for the first time, utilizes unbiased immune and metabolic transcriptome analysis of peripheral blood mononuclear cells acutely after burn, inhalation, and combined burn-inhalation injuries in patients. This novel analysis defined intrinsic mTOR and NFkB-based regulatory feedback mechanisms that occur after these injury modalities. These pathways can be utilized as biomarkers to guide clinical care, and as therapeutic targets to normalize the immune and metabolic response to burn and inhalation injuries.

## Figures and Tables

**Figure 1 ijms-23-09418-f001:**
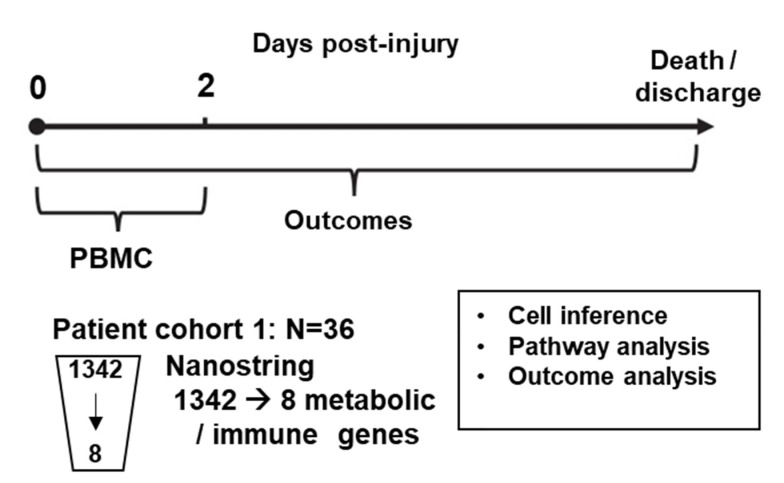
Overall experimental plan.

**Figure 2 ijms-23-09418-f002:**
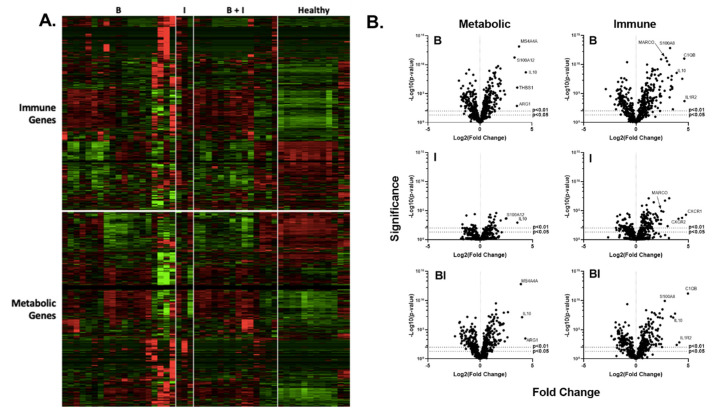
Burn injury, inhalation, and BI injuries induce changes in PBMC gene expression profiles. mRNA was isolated from PBMC, which was isolated from burn patients or healthy controls, and gene expression was evaluated using nanostring barcoding (nCounter Human Immune Profiling PanCancer CodeSet; 770 genes, and the Human Metabolic Gene CodeSet; 768 genes). Data are presented (**A**) as a heat map clustered hierarchically (green, downregulated; red, upregulated) and (**B**) as the log2-transformed differential fold change in metabolic and immune gene expression with associated *p*-value significance (using Welch’s *t*-test) after data normalization to housekeeping and internal control genes with nSolver v4.0.

**Figure 3 ijms-23-09418-f003:**
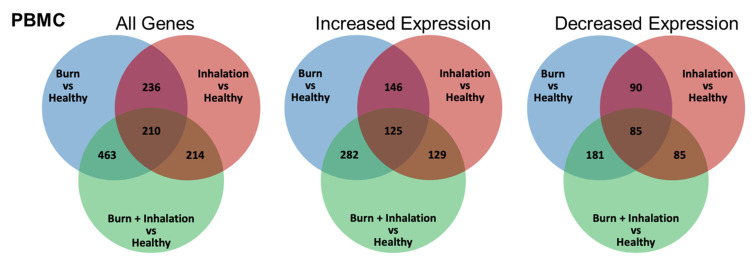
Burn injury, inhalation, and BI injuries induce unique and common gene expression profiles within PBMC. mRNA was isolated from PBMC, which was isolated from burn patients or healthy controls, and gene expression was evaluated using nanostring barcoding (nCounter Human Immune Profiling PanCancer CodeSet; 770 genes, and the Human Metabolic Gene CodeSet; 768 genes). The venn diagrams summarize the number of genes that are significantly (*p* < 0.05 using Welch’s t-test after data normalization to housekeeping and internalized the control genes with nSolver v4.0) impacted in a unique or common fashion to each injury type (when quantified *versus* healthy).

**Figure 4 ijms-23-09418-f004:**
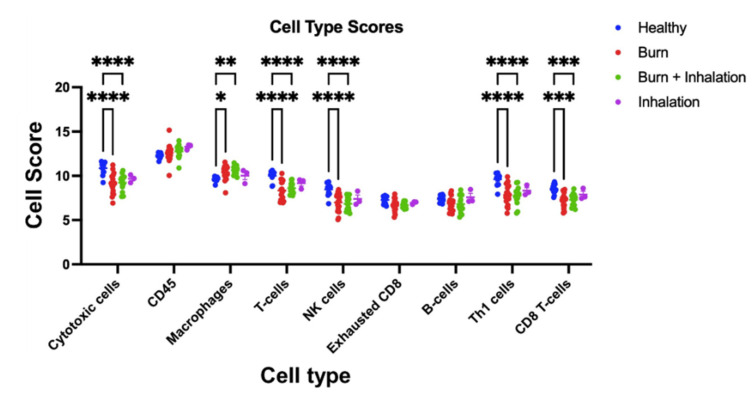
Burn injury alone and combined BI injury, but not inhalation injury alone, induce significant alterations in the composition of the peripheral immune cell populations compared to healthy controls. mRNA was isolated from PBMC which was isolated from burn patients or healthy controls and expression of immune gene mRNAs canonically associated with specific immune cell populations were analyzed in order to calculate cell type scores allowing for characterization of the cellular components within the PBMC. One-way ANOVA with Bonferroni’s multiple comparisons was used to determine significance. Data shown ± SEM; * *p* < 0.05, ** *p* < 0.01, ***, *p* < 0.005, ****, *p* < 0.001.

**Figure 5 ijms-23-09418-f005:**
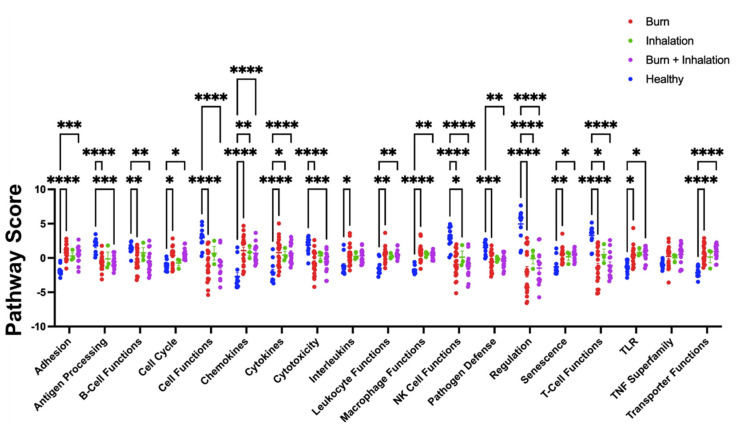
Gene expression changes following burn injury alone, inhalation injury alone, and combined BI injury result in significant different pathway scores within immunometabolic pathways. mRNA was isolated from PBMC, which was isolated from burn patients or healthy controls, and gene expression was evaluated using nanostring barcoding (nCounter Human Immune Profiling PanCancer CodeSet; 770 genes, and the Human Metabolic Gene CodeSet; 768 genes). Pathway scores (PS) from principal component analysis (PCA) of the gene expression data were generated utilizing nSolver v4.0 and R. One-way ANOVA with Bonferroni’s multiple comparisons was used to determine significance. Data shown ± SEM; * *p* < 0.05, ** *p* < 0.01, ***, *p* < 0.005, ****, *p* < 0.001.

**Figure 6 ijms-23-09418-f006:**
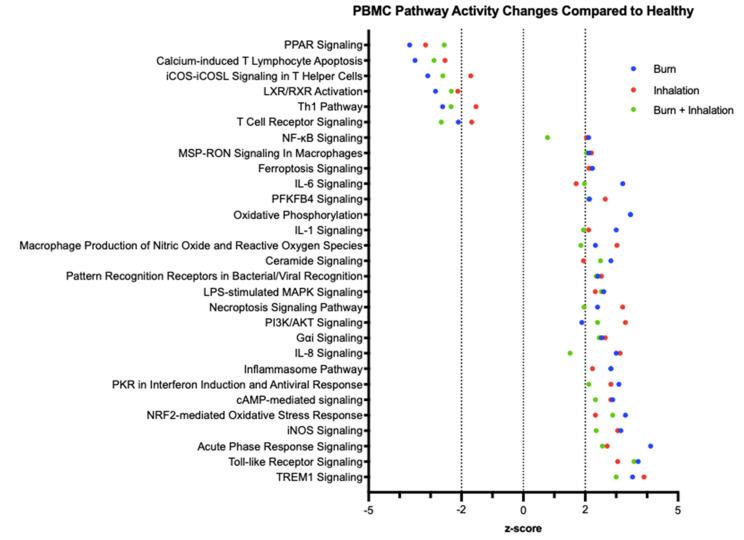
Burn, inhalation, and BI injuries induce significant alterations in peripheral immunometabolic canonical immune and metabolic signaling pathways compared to healthy controls. mRNA was isolated from PBMC, which was isolated from burn patients or healthy controls, and gene expression was evaluated using nanostring barcoding (nCounter Human Immune Profiling PanCancer CodeSet; 770 genes, and the Human Metabolic Gene CodeSet; 768 genes). Ingenuity pathway analysis (IPA) was employed to detect significant (*p* < 0.05) changes in canonical immune and metabolic pathways and quantify Z-scores.

**Figure 7 ijms-23-09418-f007:**
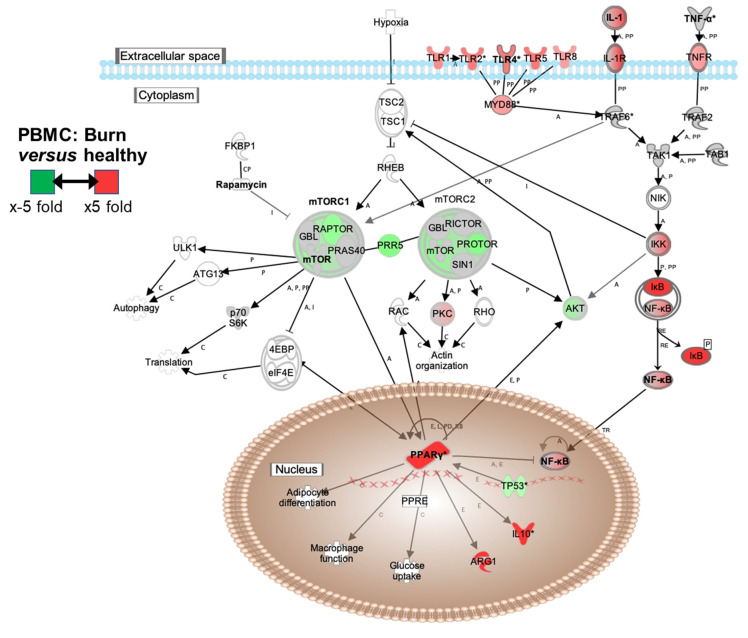
Burn injury induces significant alterations in peripheral immunometabolic canonical immune and metabolic signaling pathways compared to healthy controls. mRNA was isolated from PBMC, which was isolated from burn patients or healthy controls, and gene expression was evaluated using nanostring barcoding (nCounter Human Immune Profiling PanCancer CodeSet; 770 genes, and the Human Metabolic Gene CodeSet; 768 genes). Fold changes were generated utilizing nSolver v4.0 and R, and significantly altered (*p* < 0.05) gene expression data versus healthy controls were plotted on a canonical pathway schematic for specific mTOR, TLR, and NFκB pathways. Key: *, resolved expression value from duplicate entries in the dataset in Figure; A, activating relationship; C, causation; CP, chemical-protein interaction; E, expression; I, inhibition; L, molecular cleavage; P, (de)phosphorylation; PD, protein-DNA binding; PP, protein-protein binding; RB, regulation of binding.

**Figure 8 ijms-23-09418-f008:**
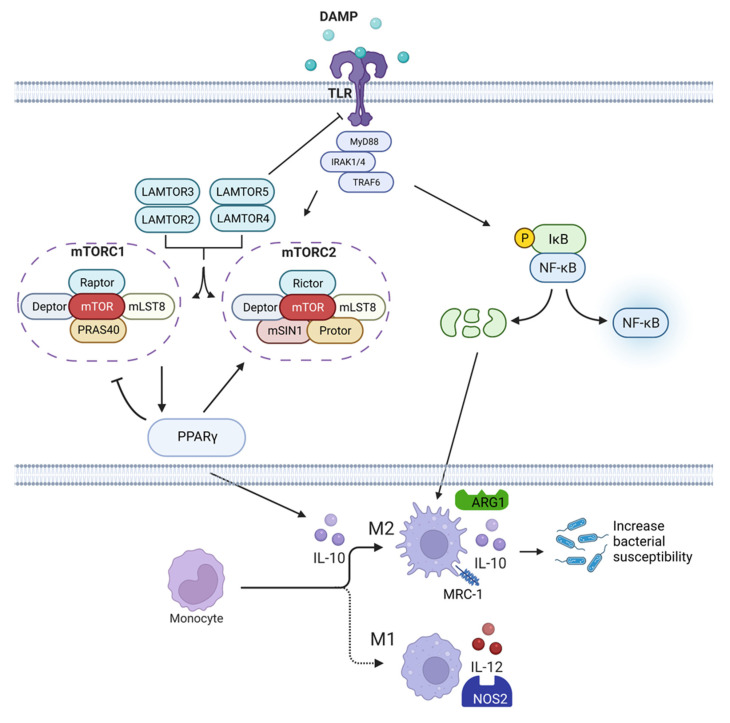
Release of damage-associated molecular patterns (DAMPs), and activation of mTOR/PPARγ and NFkB axes drive mixed antagonist response syndrome (MARS) immune dysfunction after burn and inhalation injuries (created by BioRender). Proposed molecular model of the acute burn injury-induced imbalance within the mTOR/PPARγ and NFκB/IκB immunoregulatory axes early after an injury that induces and regulates the acute shock → chronic clinical susceptibility to infection.

**Table 1 ijms-23-09418-t001:** Patient demographics.

	Injury		
	Burn	Inhalation	Burn and Inhalation
N	19	3	14
Age, mean (SD), (yrs)	44.7 ± 16.5	47.8 ± 22.7	58.8 ± 12.6
Sex (% Female)	6 (32%)	2 (67%)	8 (57%)
Race (% White)	11 (58%)	2 (67%)	9 (64%)
BMI, mean (SD)	35.2 ± 12.3	33.9 ± 13.4	28.7 ± 9.6
Total Burn Surface Area (%), mean (SD)	21.4 ± 11.2	N/A	20.3 ± 21.9
Developed Acute Lung Injury	3 (16%)	2 (67%)	11 (79%)
Require Ventilator	8 (42%)	3 (100%)	14 (100%)
Developed Infection	6 (32%)	2 (67%)	11 (79%)
Length of Stay, mean (SD), (days)	33.4 ± 27.0	11.3 ± 1.5	91.4 ± 159.5
Mortality	1 (5%)	0 (0%)	3 (21%)

**Table 2 ijms-23-09418-t002:** Linear regression modeling parameter estimates, standard error and *p*-values for outcomes of interest. Significance defined as * = 0.05 < *p* < 0.1, ** = *p* < 0.05.

	DEPTOR (Est/SE)	LAMTOR5 (Est/SE)	MRC1 (Est/SE)	PPARG (Est/SE)	RPTOR (Est/SE)
Intercept	0.412/0.356	−0.088/0.535	0.526/1.247	−0.604/1.168	−0.034/0.506
Age	−0.007/0.004	0.001/0.006	−0.005/0.015	0.012/0.014	0.001/0.006
Sex (Male)	0.084/0.141	−0.264/0.212	0.181/0.495	−0.196/0.464	0.222/0.201
Race (Non-White)	−0.187/0.135	0.071/0.203	−0.380/0.473	−0.180/0.443	−0.132/0.192
BMI	**0.013/0.006 ****	**0.026/0.009 ****	0.011/0.022	**0.050/0.021 ****	**−0.023/0.009 ****
% TBSA	0.006/0.006	0.006/0.009	−0.028/0.020	−0.009/0.019	−0.002/0.008
Inhalation (Yes)	0.343/0.199 *	−0.023/0.299	−0.544/0.697	0.047/0.652	0.051/0.283
Length of Stay	0.000/0.001	0.000/0.001	**0.006/0.003 ****	0.003/0.002	−0.001/0.001
Expired	0.242/0.245	0.303/0.368	**1.803/0.859 ****	−0.032/0.804	**−0.652/0.348 ***
Acute Lung Injury	−0.126/0.197	−0.105/0.296	−0.162/0.690	−0.584/0.646	0.170/0.280
Positive Blood or BAL Culture	−0.033/0.164	0.032/0.246	−0.126/0.574	0.530/0.537	0.050/0.233
Required Ventilator	−0.231/0.231	−0.183/0.263	0.987/0.615	0.104/0.575	0.026/0.249

## Data Availability

The data presented in this study are openly available in the Appendix A.

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
