# Peer review of "Multiplexed Human Gene Expression Analysis Reveals a Central Role of the TLR/mTOR/PPARγ and NFkB Axes in Burn and Inhalation Injury-Induced Changes in Systemic Immunometabolism and Long-Term Patient Outcomes"

_ijms, 2022, doi:10.3390/ijms23169418_

Round 1

Reviewer 1 Report

A generally competent and well-written paper. I look forward to seeing this paper in circulation. A minor suggestion: consider having a conclusion section to tie up the main findings of the paper in a succinct manner.

Reviewer 2 Report

This manuscript needs to be improved before it can be considered for publication.

Improve engilsh grammar (avoid the use of "we",  etc) and paper organization (intro, theory/analysis/process, results, conclusions....some sections are missing and the order is incorrect)

--make sure all variables and abbreviations are defined before their use or include a nomenclature.

--include a good discussion of the novel aspects of the manuscript

--avoid lumping the references...each needs to be discussed individually 

--all equations should not be in paragraphs, they should have equation numbers associated with each equation  (line 275, etc)

--axes titles need to have proper spacing and titles

--detailed discussion of the sensitivity/ error of the analysis

--provide more current  (less than 5 yrs old) references (there are only 20/100 ref that are current) and some more MDPI references

Round 2

Reviewer 2 Report

This is an improved version of the manuscript however there are still some minor english grammar problems that still must be resolved (word choice , don't use "we", etc) and some organizational issues (placement of the various sections - conclusion should be at the end right after the results, etc)

Round 3

Reviewer 2 Report

This improved version of the manuscript has addressed most of the concerns of this reviewer. 

It is recommended that in the future that the authors highlight the changes to make it easier for the reviewers to look at changes.